# Mining Highly Visited Co-Location Patterns Based on Minimum Visitor Similarity Constraints

**Xiaoxuan Wang [1], Peijie Jin [1], Wen Xiong [1],\* and Song Gao [2]**

1 School of Information Science and Technology, Yunnan Normal University, Kunming 650504, China; wangxiaoxuan1037@163.com (X.W.); jinpeijie_j@126.com (P.J.)
2 National Pilot School of Software, Yunnan University, Kunming 650504, China; gaoyitu@foxmail.com
\* Correspondence: wen.xiong@ynnu.edu.cn

**Abstract:** Spatial co-location pattern is a subset of spatial features which shows association relationships based on the spatial neighborhoods. Because the previous prevalence measurements of a co-location pattern have not considered the visited information of spatial instances, co-location patterns do not reflect the social connections (such as their spatial instances are constantly visited by common or similar moving objects) between spatial features. In this paper, a special type of co-location pattern, "Highly visited co-location patterns", is proposed, which considers the spatial proximity and visitor similarity of spatial features at the same time. A new measurement, "Minimum visitor similarity", has been proposed to reflect the visitor similarity of co-location patterns. By discussing the properties of the minimum visitor similarity, we propose an efficient algorithm to mine the highly visited co-locations and give two pruning strategies to improve the efficiency of the algorithm. Finally, extensive experiments on YELP and Foursquare datasets prove the practicability and efficiency of the proposed algorithm, and we define a "Social Entropy" to prove that spatial features in the co-locations we mined have stronger social connections.

**Keywords:** spatial data mining; co-location patterns; visitor similarity; visiting information





## 1. Introduction

As one of spatial knowledge discovery tasks, spatial co-location pattern mining aims to discover correlations between spatial features. A co-location pattern represents a subset of spatial features whose instances are frequently located together in spatial neighborhoods. The feature set {Hospital, Drugstore} may be a real-world example of co-locations since they are frequently located near each other. Spatial co-location pattern mining is defined as: Given a spatial feature set $F = \{f_1, f_2, \ldots, f_n\}$ and a spatial instances set $O = \{o_1, o_2, \ldots, o_m\}$. Each spatial instance $o_i$ belongs to a spatial feature and consists of the following fields: *instance id, location,* and *spatial feature,* where *spatial feature* $\in F$. The problem of discovering spatial co-location patterns is to find a set of spatial features $c = \{f_1, f_2, \ldots, f_k\}$ whose instances tend to locate together in close spatial regions, and $\{f_1, f_2, \ldots, f_k\} \subseteq F$. Co-location pattern mining usually uses a support measure ("Participation Index") to evaluate how frequently the features in a co-location pattern are located closely. Given a co-location pattern $c = \{f_1, f_2, \ldots, f_k\}$, a participation index $PI(c)$ is defined as $min_{f_i \in c}\{PR(c, f_i)\}$, where $PR(c, f_i)$ is the participation ratio of spatial feature $f_i$ in the pattern $c$. The Participation Ratio (PR), $PR(c, f_i)$, is calculated by $PR(c, f_i) = |N(c, f_i)| / |N(f_i)|$, $|N(c, f_i)|$ denotes the number of spatial instances of feature $f_i$ that appear in the clique instances of pattern $cp$ (a clique instance of $c$ is a group of neighbor instances whose spatial features are included in $c$), and $|N(f_i)|$ denotes the number of instances of spatial feature $f_i$ in the spatial data set. Given a user-specified threshold of participation index *min_prev*, if PI($c$) ≥ *min_prev*, co-location $c$ is a prevalent pattern in the spatial dataset. As seen from [1], the PI-based mining algorithm satisfies "Downward closure". For example, a spatial dataset $S$ is shown in Figure 1a, the

clique instances of co-location {A, B} is {{A.1,B.1},{A.2,B.1},{A.3,B.2},{A.4,B.3}}, $PR$({A, B}, A) = 4/4 = 1, $PR$({A, B}, B) = 3/3 = 1, $PI$({A, B}) = 1. If we set *min_prev* = 0.6, {A, B} must be a prevalent co-location pattern.

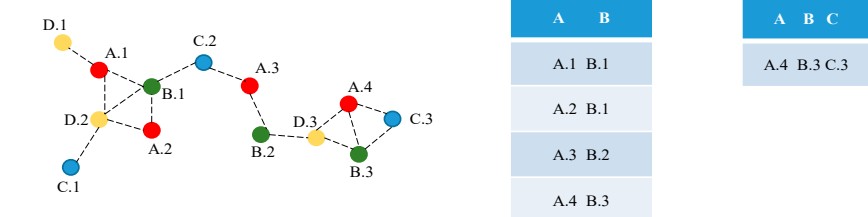

(**a**) Table instance of co-location pattern {A, B} and {A, B, C}

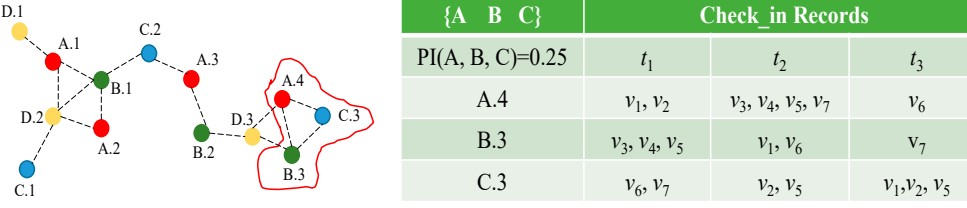

| {A  B  C} | Check_in Records | | |
|---|---|---|---|
| | $t_1$ | $t_2$ | $t_3$ |
| PI(A, B, C)=0.25 | | | |
| A.4 | $v_1, v_2$ | $v_3, v_4, v_5, v_7$ | $v_6$ |
| B.3 | $v_3, v_4, v_5$ | $v_1, v_6$ | $v_7$ |
| C.3 | $v_6, v_7$ | $v_2, v_5$ | $v_1, v_2, v_5$ |

(**b**) Check_in records of co-location pattern {A, B, C}

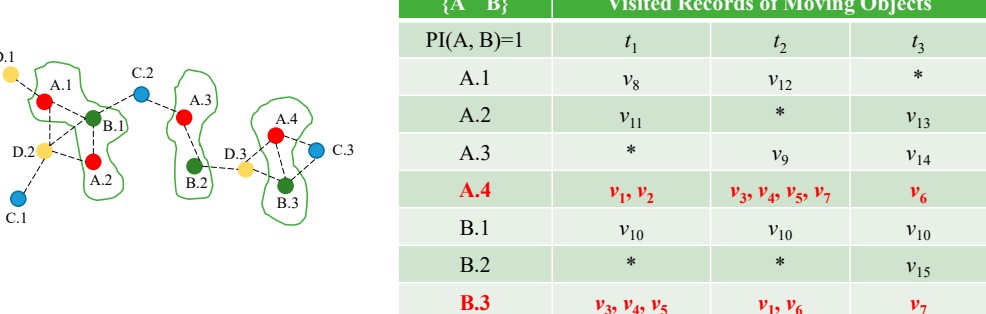

| {A  B} | Visited Records of Moving Objects | | |
|---|---|---|---|
| | $t_1$ | $t_2$ | $t_3$ |
| PI(A, B)=1 | | | |
| A.1 | $v_8$ | $v_{12}$ | * |
| A.2 | $v_{11}$ | * | $v_{13}$ |
| A.3 | * | $v_9$ | $v_{14}$ |
| **A.4** | $v_1, v_2$ | $v_3, v_4, v_5, v_7$ | $v_6$ |
| B.1 | $v_{10}$ | $v_{10}$ | $v_{10}$ |
| B.2 | * | * | $v_{15}$ |
| **B.3** | $v_3, v_4, v_5$ | $v_1, v_6$ | $v_7$ |

(**c**) Visited records of co-location pattern {A, B}

**Figure 1.** A spatial dataset with visited information *S*. (* represents that there are no moving objects visited this instance $o_j$ at time point $t_k$).

With the increasing popularity of smartphone devices, mobile users can share their check-in locations with friends (such as restaurants and theaters), which allows spatial instances to contain information that users have visited. For example, Figure 1b is the check-in records of the co-location pattern {A, B, C}, the clique instances {A.4, B.3, C.4} satisfy spatial neighbor relationship. Among them, A.4 is visited by the moving objects $v_1$ and $v_2$ at time point $t_1$, and at different time points, the moving objects are also different. Compared with traditional spatial datasets, the check-in records of these users can enrich the association relationships between spatial instances and bridge the gap between real spatial geographic networks and virtual social networks. The geographical proximity relationship between spatial instances can be seen as static, while check-in records are dynamic visited information. Additionally, the similarity of moving objects in different moments reflects the comprehensive attractiveness of several instances to users. This extraordinary attractiveness to users indicates that the distance relationship and functional combination between instances have reached a reasonable state, which can better serve users.

Unfortunately, traditional co-location does not consider the visited information of the spatial instance. By analyzing the results of existing mining methods, we can find that some mined co-location patterns are difficult to explain why their instances are always adjacent. For example, in the spatial co-location pattern <barber shops, restaurants>, people tend to build barber shops and restaurants in adjacent locations. If only considering the spatial distance, it seems difficult to explain why barbershops and restaurants are

often adjacent geographically. In fact, after analyzing the users who visited them, we found that barber shops and restaurants always have similar visitors. Consumers always like to eat at nearby restaurants after haircuts, and these consumers always have similar consumption needs. To provide convenient services to consumers, a spatial co-location pattern <barber shops, restaurants> has gradually emerged in urban development. It can be seen that when mining spatial co-location patterns, considering the similarity of visited information of spatial instances, the spatial co-location patterns mined will have more semantic information, making it easier to understand.

Based on the above analysis, this paper considers the spatial co-location pattern mining problem based on visited information. We called this special co-location pattern the ***Spatial Highly Visited Co-location Pattern*** **(SHVCP)**. SHVCP considers both the spatial proximity and social similarity between spatial instances, with the aim of mining more reasonable and easily understandable spatial co-locations.

However, a spatial highly visited co-location pattern does not necessarily mean it is also frequent. The traditional frequent spatial co-location pattern mining method only considers spatial relationships. It does not reflect the visited similarity between spatial instances, resulting in the omission of some co-location patterns with high visited similarity but not frequent during the mining process. For example, Figure 1a shows the table instances of co-location {A, B} and {A, B, C}. Assuming the participation threshold *min_prev* = 0.5, pattern {A, B} is a frequently co-located pattern with PI({A, B}) = 1 > 0.5, while {A, B, C} is not frequently co-location pattern with PI({A, B, C}) = 0.25 < 0.5. According to Figure 1a, it can be obtained that the participating instances of {A, B} are {A.1, A.2, A.3, A.4, B.1, B.2, B.3}, and the participating instances of {A, B, C} are {A.4, B.3, C.3}. Next, we analyzed the visited information of each participating instance at three consecutive times $t_1$, $t_2$, and $t_3$. We found that {A.4, B.3, C.3} were all visited by the moving object {$v_1$, $v_2$, $v_3$, $v_4$, $v_5$, $v_6$, $v_7$} at different times. Among the seven instances {A.1, A.2, A.3, A.4, B.1, B.2, B.3}, except for neighboring pairs {A.4, B.3} that have been visited by similar or same instances, there is not much similarity in the moving objects between other instances that satisfy spatial proximity. For example, in neighboring pairs {A.1, B.1}, A.1 have been visited by moving $v_8$ and $v_{12}$, while the moving object in B.1 is $v_{10}$. Co-location {A, B, C} are not frequent patterns, and their frequency is much lower than patterns {A, B}, but the similarity of moving objects is much greater than patterns {A, B}. According to the similarity of the visiting users, it can be seen that the attractiveness of {A, B, C} to users is higher than that of {A, B}. Unfortunately, traditional mining methods based on participation index only consider the spatial neighbor relationships, so they cannot discover patterns like {A, B, C}.

Due to the consideration of visited information, a highly visited co-location pattern has its particularity in both pattern measurement and mining algorithms. It is a very thought-provoking issue to define a measurement method that can reflect both the spatial proximity and visited similarity between spatial features. To better evaluate the visited similarity of co-location patterns and make up for the omission of traditional mining methods, this paper proposes a new metric to measure spatial co-location patterns, called "***Minimum Visited Co-location Similarity***", by combining the visited similarity between spatial instances after building the spatial proximity relationship in the spatial hierarchy. Summing up, the contributions of this paper are as follows:

(a) Combining the theory of visited similarity with spatial co-location pattern mining, the problem of mining high visited spatial co-location patterns is proposed and formulated.

(b) An effective visitor similarity measure of the spatial neighborhood is proposed, and the visitor similarity between spatial features in a co-location is also given.

(c) A new mining framework based on visitor similarity is designed for mining the highly visited co-location patterns, and two pruning strategies are presented to improve the efficiency of the mining algorithm.

(d) In the experimental evaluation, the differences between minimum visitor similarity and participation index are analyzed clearly. At the same time, the social entropy

is defined to compare the social connections of co-location, and extensive experiments on three real-world data sets prove the practicability and efficiency of the proposed algorithm.

The remainder of this paper is organized as follows. Section 2 presents related works of spatial co-location mining and social network analysis and mining. Section 3 discusses the measure index of highly visited co-locations. Section 4 is a basic mining algorithm and two efficient pruning strategies. Section 5 evaluates the performance of our algorithm using three real data sets. Section 6 presents a conclusion and suggests future work.

## 2. Related Work

**The spatial co-location pattern mining methods.** Since Han et al. laid the foundation of frequent pattern mining from transaction databases [2], data mining has derived different kinds of research directions, such as object discovery [3], sequential data mining [4], link prediction [5], high utility items mining [6] and spatial data mining [7]. Spatial co-location pattern mining is an important research direction in spatial data mining. The problem of frequent proximity of spatial instances in geographic areas was first defined in [8]. In ref. [1], the researchers mainly focus on the mining problem of identifying groups of particular features that frequently appear close to each other in a geospatial map, and they extend the co-location pattern model by a constraint graph, which the technique is an extension of a spatial join algorithm that operates on multiple inputs and counts long pattern instances. And ref. [1] proposes the minimum participation ratio, i.e., participation index (PI), as an interesting measure of spatial co-locations. Because PI satisfies the "Downward Closure Property", it can effectively mine co-location patterns by Apriori-like methods. Many PI-based co-location pattern mining algorithms have been proposed, such as the Join-based algorithm based on join operations [1] and the Join-less algorithm beads on star neighbors [9]. To better evaluate the proximity and direction between two points [10], measure the neighborhood relationship between spatial instances based on density weights to mine different co-location patterns. In addition, considering the sensitivity of mining results to PI [11] addresses the problem of mining confident co-location rules without a support threshold. The condensed-tree-based strategy mines the maximal co-location patterns, and a fast spatial-saving algorithm is proposed in ref. [12] to improve the efficiency of maximal co-location pattern mining. Considering the importance of uncertain data [13], it mainly studies the co-location mining problem in uncertain data. It proposes the definition of probabilistic prevalent co-locations to find all the co-locations that are likely to be prevalent in a randomly generated possible world. The non-redundant co-location patterns are defined in ref. [14] based on the concept of semantic distance between patterns. Ref. [15] uses the MapReduce framework to mine co-location patterns in parallel to cope with the huge amount of spatial data. Ref. [16] proposes parallel grid-based co-location mining algorithms on GPUs. To effectively reduce the computational complexity and improve the efficiency of spatial pattern mining, the efficiency of the original join-less algorithm is improved by querying cliques [17]. And ref. [18] introduces a range query to improve the mining efficiency. Ref. [19] shows that existing support measures suffer from various weaknesses, and thus, the authors propose a new measure called Fraction-Score, which quantifies the prevalence of candidates properly.

In addition to the above PI-based mining algorithms, researchers also mine co-location patterns from the perspective of clustering [20] and statistics [21]. The relationship between pattern mining and clustering is closely related, so ref. [20] proposes a novel framework for co-location mining using clustering techniques and shows that clustering techniques can be applied to reveal the rich structure formed by co-located spatial features. At the same time, ref. [21] designs an algorithm for finding co-location and segregation patterns based on a statistical test. And ref. [22] focus on the extended objects and design an efficient algorithm.

However, it is noteworthy that the above research is all based on static spatial data during the mining process without considering the accessed information of spatial instances.

Therefore, as an extension of co-location pattern mining, the mining of SHVCP in this paper is worth noting and researching.

**The analysis and mining methods are based on social connections.** The study of social connections originated in the social science, anthropology, and business communities. In recent years, the development of advanced techniques for Social Network Analysis and Mining (SNAM) has been highly influenced by the internet, the social web, and other large-scale socio-technical infrastructures, which are widely analyzed using graph theory, statistics, and data mining and machine learning techniques. People perceive the Web increasingly as a social medium that fosters interaction among people, sharing of experiences and knowledge, group activities, and community formation and evolution. These trends have led to the rising prominence of SNAM in academia, politics, homeland security, and business. In ref. [23], the authors conducted a sociometric study which examined the friendship network among 270 city residents, and an inverse relationship was found between similarity of friends and proximity of residence. The discovery and analysis of community structure in networks is a topic of considerable recent interest within the physics community and ref. [24] proposes a hierarchical agglomeration algorithm for detecting community structure. With the continuous development of location-based social networks, geography and social relationships are inextricably intertwined. Many studies combine geographic location information with user's social information to better analyze my social networks. Ref. [25] measures the relationship between geography and friendship and introduces an algorithm that predicts the location of an individual from a sparse set of located users with performance that exceeds IP-based geolocation. Based on the historical check-in data, the temporal and spatial impacts of crowd check-in behavior are analyzed in [26], and a POI recommendation framework is proposed, which integrates spatial and temporal impacts. Refs. [27,28] focuses on spatial clustering in location-based social networks, considering the spatio-temporal information and the social relationships between users who visit clustered places. Ref. [29] proposes a Socio-Spatial Group Query (SSGQ) to select a group of nearby attendees with tight social relations. In ref. [30], it combines mobility and network measures and finds that the similarity between two individuals' movements strongly correlates with their proximity in the social network. Given the complexity of social networks, three kinds of social relationships (face-to-face social relationships, online social relationships, and self-report social relationships) are discussed in [31]. In recent years, to protect users' privacy in the discovery process, ref. [32] detects the social ties by a novel semantic-tree-based algorithm when only obscured trajectories are provided.

Since current applications are driven by knowledge [33], knowledge-driven information on geographic space and social relationships is very important. Combining spatial information and social connections can lead to more interesting knowledge in the real world. Although some studies [27–29] have considered spatio-temporal information in data analysis, there is no research that considers social connections in mining co-location patterns. In this paper, we will take visitor similarity into account when mining the co-location patterns from spatial datasets with visited information.

## 3. Definitions

In the spatial dataset with check-in records, spatial relationships between them will not change for a long time, but the moving objects to visit these spatial instances are constantly changing. One objective of this paper is to find "High visited Co-location patterns" from spatial datasets with visited information, which are spatially adjacent and highly visited by similar visitors. For example, the co-location pattern {Market, City Park} is always visited by the older men, and the co-location {Beauty snap, Active life, Chinese Restaurant} is always visited by the young girls. Highly visited co-locations are no longer simple spatial proximity, but the similarity of visitors makes these spatial adjacent points get more social information and social connections. The specific definition description is as follows.

Given a spatial dataset containing visited information $S$, $V = \{v_1, v_2, \ldots, v_n\}$ is the set of all moving objects in the dataset, $O = \{o_1, o_2, \ldots, o_m\}$ is the spatial instance set, $F = \{f_1, f_2, \ldots, f_g\}$ is the spatial feature set, $T = \{t_1, t_2, \ldots, t_h\}$ is the time domain, where each $t_k$ is a time point. The check-in record $CK = \{\langle o_i, v_j, t_k \rangle \mid o_i \in O, v_j \in V, t_k \in T\}$, which contains all check-in records of the moving object in $V$. For moving object $v_i$, its check-in location can be considered as a spatial instance set $VH(v_i) = \{o_j \mid o_j \in O \wedge \langle o_i, v_j, t_k \rangle \in CK\}$, which records all spatial instances visited by $v_i$ at different times. For example, in Figure 1, the check-in location set $VH(v_1)$ of moving object $v_1 = \{A.4, B.3, C.3\}$. In addition, based on the check-in records of all moving objects, the visitor set of spatial instance $o_i$ at time point $t_k$ can be represented as $V_{oi}(t_k) = \{v_j \mid v_j \in V \wedge \langle o_i, v_j, t_k \rangle \in CK\}$. In Figure 1, $v_1$ and $v_2$ visit the spatial instance A.4 at time $t_1$, so the set of visitors at time $t_1$ is $V_{A.4}(t_1) = \{v_1, v_2\}$.

**Definition 1 (Snapshot visitor similarity).** *Given two spatial instances $o_i$ and $o_j$, the visitor set of $o_i$ at time point $t_k$ is $V_{oi}(t_k)$, and the visitor set of $o_j$ at time point $t_s$ is $V_{oj}(t_s)$. The similarity between $V_{oi}(t_k)$ and $V_{oj}(t_s)$ is calculated as follows:*

$$sim\left(V_{o_i}(t_k), V_{o_j}(t_s)\right) = \frac{\sum_{v_a \in V_{o_i}(t_k), v_b \in V_{o_j}(t_s)} w_{v_a, v_b}}{x \cdot y} \tag{1}$$

*where $x$ is the number of moving objects in $V_{oi}(t_k)$ and $y$ is the number of moving objects in $V_{oj}(t_s)$. $w_{va,vb}$ is the cosine similarity between the moving object $v_a$ and $v_b$. In this paper, we choose the cosine similarity proposed in [5] to evaluate the similarity between two moving objects. The specific calculation method for $w_{va,vb}$ is as follows:*

$$w_{v_a, v_b} = \frac{\sum_{o_i \in O} c_{v_a, o_i} c_{v_b, o_i}}{\sqrt{\sum_{o_i \in O} c_{v_a, o_i}{}^2} \sqrt{\sum_{o_i \in O} c_{v_b, o_i}{}^2}} \tag{2}$$

*where each moving object is represented by a binary visiting vector over all $O$, and $C_{vi,oj}$ takes either 1 or 0 value, we set $c_{vi,oj} = 1$, if $o_j \in VH(v_i)$, and $c_{vi,oj} = 0$ otherwise. Obviously, $sim\left(V_{o_i}(t_k), V_{o_j}(t_s)\right) = sim\left(V_{o_j}(t_s), V_{o_i}(t_k)\right)$.*

To calculate the visitor similarity between spatial instances by the visiting information, we need to consider visitor similarity at more time points. In this paper, we give a "Visiting crowd" to collect a sequence of snapshot visitors at consecutive timestamps. Then, we calculate the average similarity of snapshot visitors in visiting crowds to reflect the visitor similarity between two spatial instances.

**Definition 2 (Visiting crowd).** *For a spatial instance $o_i$, the visiting crowd of $o_i$ is a sequence of snapshot visitors at consecutive timestamps, and the crowd of $o_i$ can be denoted as $VC_{oi} = \{V_{oi}(t_1), V_{oi}(t_2), \ldots, V_{oi}(t_n)\}$.*

In this paper, the check-in records are dynamic. To better evaluate the visitor similarity between two spatial instances, we consider the time attribute in computing the similarities between the different visiting crowds. We can calculate the visitor similarity weight between spatial instances as Definition 3.

**Definition 3 (Visitor similarity weight between spatial instances).** *Given a neighborhood pair $<o_h, o_k>$, the visiting crowds of $o_h$ and $o_k$ are $VC_{oi} = \{V_{oi}(t_1), V_{oi}(t_2), \ldots, V_{oi}(t_n)\}$, $VC_{oj} = \{V_{oj}(t_1), V_{oj}(t_2), \ldots, V_{oj}(t_n)\}$, the visitor similarity between $o_i$ and $o_j$ can be calculated as:*

$$vs\left(o_i, o_j\right) = \frac{\sum_{V_{o_i}(t_a) \in VC_{o_i}} \sum_{V_{o_j}(t_b) \in VC_{o_j}, |t_a - t_b| \leq \tau} sim\left(V_{o_i}(t_a), V_{o_j}(t_b)\right)}{\sum_{V_{o_i}(t_a) \in VC_{o_i}} \sum_{V_{o_j}(t_b) \in VC_{o_j}} \mu(|t_a - t_b|)} \tag{3}$$

*where $\mu(|t_a-t_b|) = 1$ if $|t_a-t_b| \leq \tau$, and $\mu(|t_a-t_b|) = 0$ otherwise ($\tau$ is the time threshold), $vs(o_i, o_j)$ is a number greater than 0 but less than 1, i.e., $0 \leq vs(o_i, o_j) \leq 1$. For example, the Visiting Crown of A.4 and B.3 in Figure 1 is shown in Figure 2. Making $\tau = 1$, the visitor similarity between A.4 and B.3 can be calculated as $vs(A.4, B.3) = (sim(V_{A.4}(t_1), V_{B.3}(t_1)) + sim(V_{A.4}(t_2), V_{B.3}(t_2)) + \ldots + sim(V_{A.4}(t_3), V_{B.3}(t_3)))/7$.*

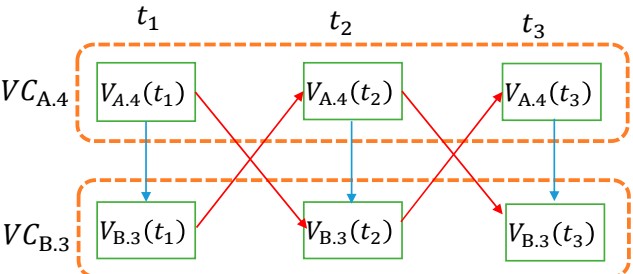

**Figure 2.** An example of visiting crowds.

For a spatial co-location $c = \{f_1, f_2, \ldots, f_k\}$, its table-instance $insT_c$ can be generated by the spatial neighborhood relationships, and the visitor similarities of these spatial neighborhoods are also calculated by Definition 3. Thus, we can analyze the social connection between features in $c$ based on visitor similarity weights of neighborhoods and $insT_c$.

**Definition 4 (Visitor similarity weight between spatial features).** *Spatial features $f_i$ and $f_j$ are two features in the co-location pattern $c = \{f_1, f_2, \ldots, f_k\}$, and $insT_c$ is the table instance of $c$, the visitor similarity weight between $f_i$ and $f_j$ can be calculated as*

$$VS_c(f_i, f_j) = \frac{\sum_{\langle o_x, o_y \rangle \in \prod_{\langle f_i, f_j \rangle} insT_c} vs(o_x, o_y)}{|insT_c|} \qquad (f_i, f_j \in c \wedge f_j > f_i) \qquad (4)$$

*where $\prod$ is a project operation, $\prod_{\langle fi, fj \rangle} insT_c$ contains all instances where features $f_i$ and $f_j$ participate in $c$, and $\langle o_x, o_y \rangle$ is a pair of spatial neighbor relationships in $insT_c$ which the spatial feature of the $o_x$ is $f_i$. The spatial feature of $o_y$ is $f_j$. The visitor similarity weight between spatial features reflects the average value of visitor similarity weight of spatial neighbor relationships between $f_i$ and $f_j$. As shown in Figure 3, the table instance length for pattern $\{A, B\}$ is 4, and the neighboring pairs of features A and B participating in patterns $\{A, B\}$ is $\{A.1, B.1\}$, $\{A.2, B.1\}$, $\{A.3, B.2\}$, $\{A.4, B.3\}$. Therefore, the visitor similarity weight between feature A and feature B in patterns $\{A, B\}$ is $VS_{\{A,B\}}(A, B) = (vs(A.1, B.1) + vs(A.2, B.1) + vs(A.3, B.2) + vs(A.4, B.3))/4 = 0.12$.*

**Definition 5 (Minimum Visitor similarity of co-location).** *Given a co-location pattern $c = \{f_1, f_2, \ldots, f_k\}$, the visitor similarity of $c$ can be calculated as:*

$$Mvs(c) = min\{VS_c(f_i, f_j) | f_i, f_j \in c \wedge f_j > f_i\} \qquad (5)$$

As shown in Figure 3, the minimum visitor similarity of $\{A, B, C\}$ is $Mvs(\{A, B, C\}) = min\{VS_{\{A,B,C\}}(A, B), VS_{\{A,B,C\}}(A, C), VS_{\{A,B,C\}}(B, C)\} = 0.33$.

Minimum visitor similarity (Mvs) is an interesting new metric based on visited information constraints. Unlike the counting rule of PI, Mvs counts the participating neighboring pairs and calculates the average visitor similarity of neighboring pairs in a co-location pattern. Normally, we set a threshold *min_sim* to discover interesting patterns and to avoid blindly setting thresholds. We need to analyze the range of values for the minimum visitor similarity. Hence, we conducted the following analysis of the properties of Mvs:

**Property 1 (Non-negative boundedness).** *The minimum visitor similarity of spatial co-location pattern $c$ satisfies non-negative boundedness, i.e., $0 \leq Mvs(c) \leq 1$.*

**Proof of Property 1.** The visitor similarity weight between the neighborhood is greater than 0 but less than 1. Therefore, based on the definition of visitor similarity between features, it can be concluded that $\sum_{\langle o_x, o_y \rangle \in \prod_{\langle f_i, f_j \rangle} insT_c} vs(o_x, o_y) \leq |insT_c|$. $VS_c(f_i, f_j)$ also satisfies the condition of greater than 0 but less than 1, thus proving that $0 \leq Mvs(c) \leq 1$. The minimum visitor similarity of spatial co-location pattern $c$ satisfies non-negative boundedness. □

**Property 2 (Does not satisfy the "downward closure" property).** *The minimum visitor similarity of co-locations does not satisfy the "downward closure" property, i.e., Mvs(c) may not necessarily be greater than Mvs(c'), c' is a super pattern of c.*

**Proof of Property 2** . Here we prove this by citing counterexamples. Figure 3 shows {A, B, C} is a super pattern of {A, B}. The minimum visitor similarity of {A, B} is $Mvs(\{A, B\}) = 0.12$, while $Mvs(\{A, B, C\}) = 0.33 > Mvs(\{A, B\})$. The minimum visitor similarity of co-locations does not satisfy the "downward closure" property. □

**Definition 6 (Spatial Highly Visited Co-location Patterns).** *According to property I, we set a threshold min_sim within the value range of [0, 1]. For a co-location pattern c, if Mvs(c) $\geq$ min_sim, then c is a spatial highly visited co-location pattern.*

**Problem Formulation: Spatial Co-location Patterns Mining based on Instance-visited Constraint**. Given a spatial dataset containing visited information S, $V = \{v_1, v_2, \ldots, v_n\}$ is the set of all moving objects in the dataset, $O = \{o_1, o_2, \ldots, o_m\}$ is the spatial instance set, $F = \{f_1, f_2, \ldots, f_g\}$ is the spatial feature set, $T = \{t_1, t_2, \ldots, t_h\}$ is the time domain, where each $t_k$ is a time point. The check-in record $CK = \{\langle o_i, v_j, t_k \rangle \mid o_i \in O, v_j \in V, t_k \in T\}$, which contains all check-in records of the moving object in V. Given a distance threshold $d$ and the visitor similarity threshold *min_sim*, the spatial high visited co-location pattern mining based on minimum visitor similarity aims to mine the co-location pattern set SCPs from the dataset S', where instances are spatially adjacent to each other (Euclidean distance is greater than the distance threshold $d$). Features have high visitor similarity, SCPs = $\{c_1, c_2, \ldots, c_h\}$ where $\forall c_i \in SCPs, Mvs(c) \geq min\_sim$.

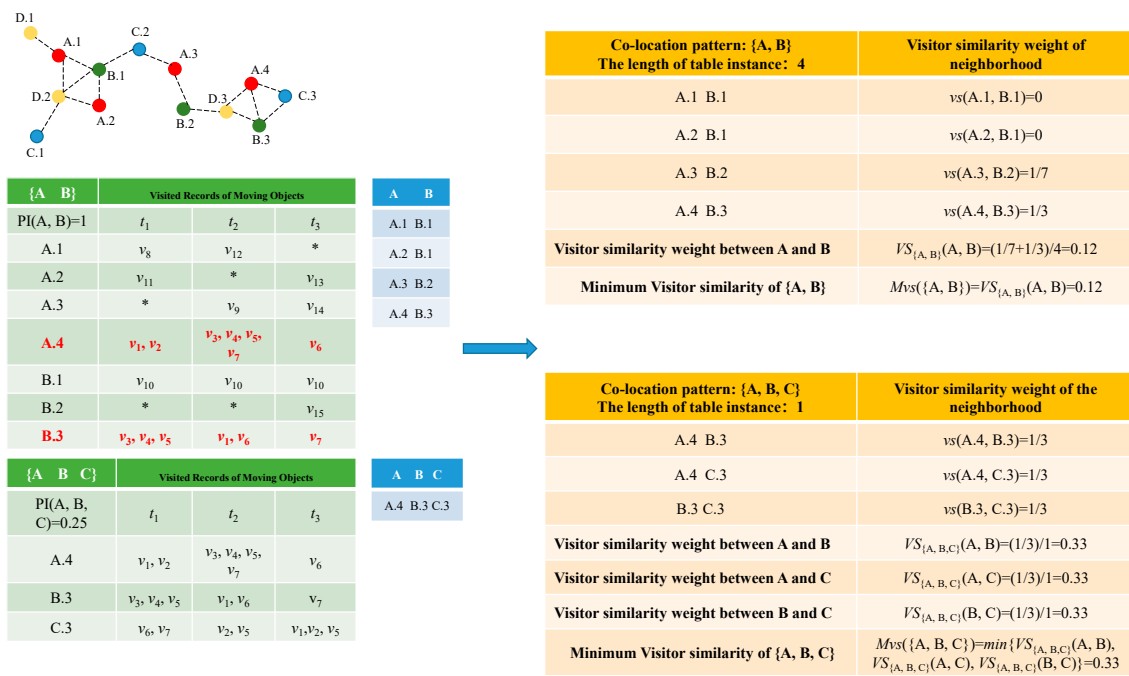

**Figure 3.** Minimum Visitor similarity of co-location patterns. (* represents that there are no moving objects visited this instance $o_j$ at time point $t_k$).

## 4. Algorithms and Analysis

In this section, we discuss the process of mining spatial high visited co-location patterns. The previous section has proven that the minimum visitor similarity does not meet the property of "downward closure". Therefore, it is necessary to propose an effective mining algorithm to compensate for the increased computational complexity. This section introduces a basic algorithm and two optimization strategies and provides a detailed analysis of the complexity of the algorithm.

### 4.1. Baseline Algorithm

Firstly, this section proposes a baseline algorithm (Algorithm 1). To reduce the mining process's computational complexity and time cost, we introduced a neighbor instance list of row instances [19] to accelerate the generation of candidate and pattern table instances. We can automatically generate co-located spatial features without generating non-clique candidates at each level. For the spatial instance set $O = \{o_1, o_2, \ldots, o_m\}$, the neighbor instance list of $o_i$ can be denoted as $NIL(o_i) = \{o_j \mid o_j \in O \wedge dist(o_i, o_j) \leq d \wedge f_{oj} > f_{oi}\}$ which means any spatial instance in $NIL(o_i)$ has a neighbor relationship with $o_i$. Given a set of neighbor instance list $\{NIL(o_1), NIL(o_2), \ldots, NIL(o_m)\}$, assuming that $<o_1, o_2, \ldots, o_k>$ is a row instance of co-location $c$, the can be calculated as $NIL(o_1) \cap NIL(o_2)$ and the $NIL(<o_1, o_2, \ldots, o_k>)$ can be calculated as $NIL(<o_1, o_2, \ldots, o_{k-1}>) \cap NIL(o_k)$, in other words, $NIL(<o_1, o_2, \ldots, o_k>) = NIL(o_1) \cap NIL(o_2) \cap \ldots \cap NIL(o_k)$. By the neighbor instance list of a subset, we can quickly generate the co-location candidates and their table instances. Based on the neighbor instance list, the idea of the basic algorithm is very simple. BA generates all candidate patterns step by step, then calculates the minimum visitor similarity for each candidate, and collects spatial high visited co-location patterns by visitor similarity threshold *min_sim*. The advantage of the basic algorithm is that it does not lose any patterns, ensuring the completeness and correctness of the results.

---

**Algorithm 1.** Baseline algorithm based on minimum visitor similarity

---

**Input:** Spatial feature set $F = \{f_1, f_2, \ldots, f_g\}$, spatial instance set $O = \{o_1, o_2, \ldots, o_m\}$, moving object set $V = \{v_1, v_2, \ldots, v_n\}$, time set $T = \{t_1, t_2, \ldots, t_h\}$, Check_in records $CK = \{\langle o_i, v_j, t_k \rangle \mid o_i \in O, v_j \in V, t_k \in T\}$, distance threshold $d$, time threshold $\tau$, minimum visitor similarity threshold *min_sim*.
**Output:** Spatial High Visited Co-location Pattern Set (SCPs)
**Begin**
1. Calculate spatial neighborhoods table containing visitor similarity;
2. Calculate and collect 2-size spatial high visited co-location patterns;
3. Make $k = 3$, generate k-size candidate patterns according to NIL and store them in $CM_k$.
4. While $CM_k$ != NULL
    4.1 For each $c \in CM_k$
        4.1.1 Generate the table instance $insT_c$ of candidate $c$ based on NIL;
        4.1.2 Calculate the minimum visitor similarity of co-location $c$ $Mvs(c)$;
        4.1.3 If $Mvs(c) \geq min\_sim$, $c$ is a spatial highly visited co-location pattern and put $c$ into SCPs;
    4.2 Generate k + 1-size candidates $CM_{k+1}$ by $CM_k$ and NIL, $k = k + 1$;
5. Put out all spatial high visited co-location patterns SCPs;
**End**

---

### 4.2. Optimization Strategy Based on Table Instance Checking

**Lemma 1.** *For the co-location c and its super set c', if c is not a high visited co-location pattern, and $|insTc| \leq |insTc'|$, c' must not be a high visited co-location pattern.*

**Proof of Lemma 1.** Given a *k*-size co-location pattern $c = \{f_1, f_2, \ldots, f_k\}$, and its super co-location pattern $c' = \{f_1, f_2, \ldots, f_k, f_{k+1}\}$, $insT_c$ is the table instances of c and $insT_{c'}$ is the table instances of $c'$. Due to $\sum_{\langle o_h, o_k \rangle \in \prod_{\langle f_i, f_j \rangle} insT_c} vs(o_h, o_k) \geq \sum_{\langle o_h, o_k \rangle \in \prod_{\langle f_i, f_j \rangle} insT_{c'}} vs(o_h, o_k)$ and $|insT_c| \leq$

$|insT_{c'}|$, we can get that $\dfrac{\sum_{\langle o_h,o_k\rangle\in\Pi_{\langle f_i,f_j\rangle}\ insT\ \ c}\ vs(o_h,o_k)}{|Tins(cp)|}\geq\dfrac{\sum_{\langle o_h,o_k\rangle\in\Pi_{\langle f_i,f_j\rangle}\ Tins(cp\ ')}\ vs(o_h,o_k)}{|Tins(cp')|}$ which means $VS_c(f_i,f_j)\geq VS_{c'}(f_i,f_j)$. So, $Mvs(c)\geq Mvs(c')$, and if co-location $c$ is not a high visited co-location, the super pattern $c'$ is also not a high visited co-location. □

Shown as the example in Figure 4, the co-location pattern {A, B} and {A, B, C}, $|insT_{\{A,B\}}|\leq|insT_{\{A,B,C\}}|=4$, we can get that $Mvs(\{A, B\})\geq Mvs(\{A, B, C\})$.

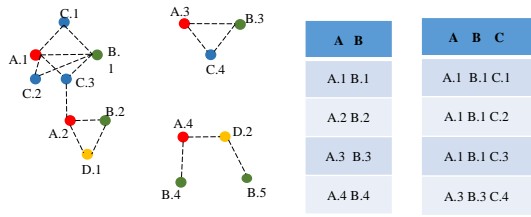

| A B |
|---|
| A.1 B.1 |
| A.2 B.2 |
| A.3 B.3 |
| A.4 B.4 |

| A B C |
|---|
| A.1 B.1 C.1 |
| A.1 B.1 C.2 |
| A.1 B.1 C.3 |
| A.3 B.3 C.4 |

| Co-location pattern: {A, B} The length of table instance: 4 | Visitor similarity weight of neighborhood |
|---|---|
| Visitor similarity weight between A and B | $VS_{\{A, B\}}(A, B)=(0.1+0.4+0.1+0.15)/4=0.1875$ |
| Minimum Visitor similarity of {A, B} | $Mvs(\{A, B\})=VS_{\{A, B\}}(A, B)=0.1875$ |

| Neighbor relationships | Visitor similarity weight of neighborhood | Neighbor relationships | Visitor similarity weight of neighborhood |
|---|---|---|---|
| <A.1, B.1> | $vs$(A.1, B.1)=0.1 | <A.2, D.1> | $vs$(A.2, D.1)=0.5 |
| <A.2, B.2> | $vs$(A.2, B.2)=0.4 | <A.4, D.2> | $vs$(A.4, D.2)=0.22 |
| <A.3, B.3> | $vs$(A.3, B.3)=0.1 | <B.1, C.1> | $vs$(B.1, C.1)=0.24 |
| <A.4, B.4> | $vs$(A.4, B.4)=0.15 | <B.1, C.2> | $vs$(B.1, C.2)=0.16 |
| <A.1, C.1> | $vs$(A.1, C.1)=0.2 | <B.1, C.3> | $vs$(B.1, C.3)=0.32 |
| <A.1, C.2> | $vs$(A.1, C.2)=0.12 | <B.3, C.4> | $vs$(B.3, C.4)=0.22 |
| <A.1, C.3> | $vs$(A.1, C.3)=0.3 | <B.2, D.1> | $vs$(B.2, D.1)=0.5 |
| <A.2, C.3> | $vs$(A.2, C.3)=0.9 | <B.5, D.2> | $vs$(B.5, D.2)=0.1 |
| <A.3, C.4> | $vs$(A.3, C.4)=0.3 | | |

| Co-location pattern: {A, B, C} The length of table instance: 4 | Visitor similarity weight of neighborhood |
|---|---|
| Visitor similarity weight between A and B | $VS_{\{A, B, C\}}(A, B)=(0.1+0.1)/4=0.05$ |
| Visitor similarity weight between A and C | $VS_{\{A, B, C\}}(A, C)=(0.2+0.12+0.3+0.3)/4=0.23$ |
| Visitor similarity weight between B and C | $VS_{\{A, B, C\}}(B, C)=(0.24+0.16+0.32+0.22)/4=0.235$ |
| Minimum Visitor similarity of {A, B, C} | $Mvs(\{A, B, C\})=min\{VS_{\{A, B, C\}}(A, B), VS_{\{A, B, C\}}(A, C), VS_{\{A, B, C\}}(B, C)\}=0.05$ |

**Figure 4.** A spatial dataset with visited information $S'$.

Additionally, in order to speed up the calculation speed, using a clique instance neighbor set can quickly obtain the number of row instances that the super pattern of $c$:

Given a $k$-size co-location pattern $c=\{f_1, f_2, \ldots, f_k\}$, the table instance of $c$ is $insT_c=\{I_1, I_2, \ldots, I_n\}$, the neighbor set of $insT_c$ is $\{NIL(I_1), NIL(I_2), \ldots, NIL(I_n)\}$, for the super pattern $c'=\{f_1, f_2, \ldots, f_k, f_{k+1}\}$, the number of row instances of $c'$ can be calculated as follow:

$$n_{c'}=\left|\bigcap_{i=1}^{n}\left(\prod_{f_{k+1}}NIL(I_i)\right)\right| \tag{6}$$

where $\prod_{f_{k+1}}NIL(I_i)$ represents the number of instances of $f_{k+1}$ in $NIL(I_i)$.

**Inference 1 .** *For a co-location pattern $c$ and its super pattern $c'$, if $n_{c'}\geq|insT_c|$ and $c$ is not a high visited co-location pattern, $c'$ must not be a high visited co-location pattern.*

**Proof of Inference 1.** According to the theory and method of generating the row instance by neighbor instance set, we can get that $n_{c'}=|insT_{c'}|$. Hence, if $c$ is not a high visited co-location pattern, $c'$ is also not a high visited co-location pattern. □

Based on Lemma 1 and Inference 1 described above, we propose a table instance checking strategy (Lemma 2) to reduce the computation of candidate patterns with low visitor similarity.

**Lemma 2.** *Given a $k$-size co-location pattern $c_1=\{f_1, f_2, \ldots, f_k\}$, $c_2=\{f_i, f_j\}$ is 2-size co-location pattern, $c_2\subseteq c_1$. Assuming that $c_2$ is not a spatial high visited co-location pattern, and $|insT_{c2}|\leq|insT_{c1}|$, $c_1$ must not be a high visited co-location pattern.*



**Proof of Lemma 2.** Since $c_2$ is a 2-size subset of $c_1$, we can get $\sum_{\langle o_h, o_k \rangle \in insT_{c_2}} vs(o_h, o_k) \geq \sum_{\langle o_h, o_k \rangle \in \prod_{\langle f_i, f_j \rangle} insT_{c_1}} vs(o_h, o_k)$. Then, according to the conditions of $|insT_{c2}| \leq |insT_{c1}|$, $\frac{\sum_{\langle o_h, o_k \rangle \in \prod_{\langle f_i, f_j \rangle} insT_{c_2}} vs(o_h, o_k)}{|insT_{c_2}|} \geq \frac{\sum_{\langle o_h, o_k \rangle \in \prod_{\langle f_i, f_j \rangle} insT_{c_1}} vs(o_h, o_k)}{|insT_{c_1}|}$ also can be obtained. It means that $VS_{c2}(f_i, f_j) \geq VS_{c1}(f_i, f_j)$ and $Mvs(c_2) \geq Mvs(c_1)$. If $c_2$ is not high visited co-location pattern, $Mvs(c_2) < min\_sim$, and $Mvs(c_1) < min\_sim$, $c_1$ is not a high visited co-location pattern. □

For the co-location pattern {A, B} shown in Table 1, the number of row instances of pattern {A, B, C} can be quickly obtained by searching for the neighbor instance list of {A, B}: $NIL(\langle A.1, B.1 \rangle) = \{C.1, C.2, C.3\}$ and $NIL(\langle A.3, B.3 \rangle) = \{C.4\}$. If {A, B} is not a spatial high visited co-location pattern, because $|insT_{\{A,B\}}| \leq |insT_{\{A,B,C\}}|$, {A, B, C} can no longer be computed.

**Table 1.** Neighbor Instance Set.

| Co-Location Pattern | Clique Instance | Visitor Similarity between Instances | Neighbor Instance Set | The Number of Neighbor Instances |
|---|---|---|---|---|
| $\langle A, B \rangle$ | $\langle A.1, B.1 \rangle$ | $vs(A.1, B.1)$ | C: C.1, C.2, C.3 | C: 4 |
|  | $\langle A.2, B.2 \rangle$ | $vs(A.2, B.2)$ | D: D.1 | D: 1 |
|  | $\langle A.3, B.3 \rangle$ | $vs(A.3, B.3)$ | C: C.4 |  |
|  | $\langle A.4, B.4 \rangle$ | $vs(A.4, B.4)$ | $\varnothing$ |  |

**Optimization Goals**: The main purpose of the optimization strategy for table instance checking is to reduce computational burden by simply and quickly obtaining the number of row instances of co-location patterns. This optimization strategy simplifies some calculations of candidates' visitor similarity in advance. The pseudocode is shown in Algorithm 2.

---

**Algorithm 2:** Optimization_Strategy_1

---

**Input:** Candidate pattern $c$, neighbor instance set *NIL*
**Output:** True or False
**Begin**
1. Calculate the 2-size subset $C_2(c)$ of pattern c based on NIL;
2. Calculate the number of row instances $n_c$ of pattern $c$;
3. If $\exists \langle f_i, f_j \rangle \in C_2(c)$ and satisfy Lemma 2, return False;
4. Else return true;
**End**

---

*4.3. Optimization Strategy Based on Visitor Similarity Checking*

We proposed an optimization strategy based on table instance checking in the previous section. In this section, we will discuss other optimization strategies from the perspective of visitor similarity. Firstly, by setting the visitor similarity threshold *min_sim*, the spatial neighbor relationship can be divided into G-set and L-set.

**G-set**: The all of neighborhoods in G-set, their visitor similarity weights of neighborhoods are equal or larger than *min_sim*, $\forall \langle o_i, o_j \rangle \in$ G-set, $vs(o_i, o_j) \geq min\_sim$.

**L-set**: The all of neighborhoods in L-set, their visitor similarity weights of neighborhoods are less than *min_sim*, $\forall \langle o_i, o_j \rangle \in$ L-set, $vs(o_i, o_j) < min\_sim$.

For example, we set *min_sim* = 0.4. The G-set and L-set in Figure 4 are shown in Table 2.

**Table 2.** The example of G-set and L-set.

| Co-Location Patterns | G-Set | L-Set | Co-Location Patterns | G-Set | L-Set |
|---|---|---|---|---|---|
| <A, B> | <A.2, B.2> | <A.1, B.1> <br> <A.3, B.3> <br> <A.4, B.4> | <B, C> | | <B.1, C.1> <br> <B.1, C.2> <br> <B.1, C.3> <br> <B.3, C.4> |
| <A, C> | <A.2, C.3> | <A.1, C.3> <br> <A.1, C.1> <br> <A.1, C.2> <br> <A.3, C.4> | <B, D> | <B.2, D.1> | <B.5, D.2> |
| <A, D> | <A.2, D.1> | <A.4, D.2> | | | |

**Lemma 3 .** *For a co-location candidate pattern c, if $\prod_{\langle f_i,f_j \rangle} insT_c$ are generated by the neighborhoods in L-set, the candidate c and its super co-locations must not be a high visited co-location pattern.*

**Proof of Lemma 3.** Given a co-location $c = \{f_1, f_2, \ldots, f_k\}$ and its super co-location $c' = \{f_1, f_2, \ldots, f_k, f_{k+1}\}$, . If all of the neighborhoods in $\prod_{\langle f_i,f_j \rangle} insT_c$ are generated by the L-set of spatial neighborhoods, $VS_c(f_i, f_j) = \frac{\sum_{\langle o_h,o_k \rangle \in \prod_{\langle f_i,f_j \rangle} insT_c} vs(o_h,o_k)}{|insT_c|}$ must less than the threshold *min_sim*. And we can get that $\prod_{\langle f_i,f_j \rangle} insT_{c'} \subseteq \prod_{\langle f_i,f_j \rangle} insT_c$ and $\frac{\sum_{\langle o_h,o_k \rangle \in \prod_{\langle f_i,f_j \rangle} insT_{c'}} vs(o_h,o_k)}{|insT_{c'}|}$ is also less than *min_sim*. Therefore, the pattern *c* and its super co-locations must not be a high visited co-location pattern. □

**Optimization Goals**: After dividing the spatial neighbor relationship into G-set and L-set based on visitor similarity, the visitor similarity of some candidates does not need to be calculated. Because their visitor similarity is significantly lower than the visitor similarity threshold *min_sim*, it helps improve the efficiency of mining algorithms. Under the condition of Lemma 3, *Mvs* satisfies partial downward closure. Therefore, for this optimization strategy, the optimization goal is to prune a portion of candidate patterns without generating their super patterns. Pseudocode is shown in Algorithm 3.

---

**Algorithm 3:** Optimization_Strategy_2

---

**Input:** Candidate pattern *c*, Candidates set $CM_k$, L-set
**Begin**
1. Calculate the 2-size subset $C_2(c)$ of pattern c based on *NIL*;
2. If $\exists <f_i, f_j> \in C_2(c)$ and satisfy Lemma 3, remove c from $CM_k$;
**End**

---

Combining the two optimization strategies introduced above, we propose an efficient mining algorithm: Efficient Spatial High Visited Co-location Pattern Mining Algorithm (ESHVCM). Pseudocode is shown in Algorithm 4.

*4.4. Algorithm Complexity Analysis*

- **The computational complexity of 2-size co-locations** generation. In this paper, the method of grid scanning is used to calculate spatial proximity relationships. Assuming every spatial instance has $n_p$ spatial instances in adjacent grids, the average number of visitors in $V_{oi}(t_k)$ of a spatial instance is $n_{user}$. According to the above definitions, calculating the spatial neighborhoods must take $mn_p$ ($mn_p \gg n_p$) time complexity (*m* is the number of spatial instances), and the maximum time complexity for calculating the visitor similarity weight of different spatial neighborhoods is $h^2 n_{user}^2$ (*h* is the number of time point). Hence, the total complexity of 2-size co-locations generation is $O(mn_p + h^2 n_{user}^2)$.

- **The computational complexity of neighbor instance list.** According to the neighbor instance list $NIL(c)$, $c = \{f_1, f_2, \ldots, f_k\}$ is a $k$-size co-location, we can quickly generate the row-instances of $(k + 1)$-size co-location $c'$, which is the super co-location of $c$. The spatial features set is $F$, and the row instances of $c$ can neighbor with the different spatial instances whose spatial features belong to the set $F$-$c$. Assuming that every row instance $I \in c$ is neighboring with $n_f$ spatial instances, and the number of row instances of $c$ is $l$. Due to generating the neighbor instance list need to use the intersection operation, the complexity of row instance generation is $l(n_f + n_f) = 2ln_f$; hence, the computational complexity of the neighbor instance list is $O(2kN_cln_f)$ ($N_c$ is the average number of any size co-location patterns, and $k$ is the maximum length of co-location patterns. In the worst case, $kN_c = 2m$).

- **The computational complexity of calculating the minimum visitor similarity.** For a k size co-location c, the complexity of calculating minimum visitor similarity is $C_k^2 l$. Assuming the average number of co-location patterns for any size is $N_c$, calculating the minimum visitor similarities of all co-location patterns should cost $O\left(C_2^2 N_c l + C_3^2 N_c l + \ldots + C_k^2 N_c l\right) = O(2kN_cl)$.

- **The computational complexity of pruning strategies.** Without considering the pruning strategies, we need to consume $O(2kN_clnf + 2kN_cl)$ ($2kN_clnf + 2kN_cl \gg mn_p + h^2n_{user}^2$) time complexity to collect all of the high visited co-locations. Assuming each size has an average of $N_p$ candidates that can be pruned, we can save $2kN_pln_f + 2kN_pl$ time complexity using two pruning strategies. The complexity strategy of table instance checking is $O(kN_c)$, and the complexity strategy of visitor similarity weight checking is $O(kN_cl)$. $2kN_cln_f + 2kN_cl \gg (l + 1)kN_c$. Hence, the pruning strategy greatly reduces the time complexity without increasing computation.

---

**Algorithm 4:** Efficient Spatial High Visited Co-location Pattern Mining Algorithm

---

**Input:** Spatial feature set $F = \{f_1, f_2, \ldots, f_g\}$, Spatial instance set $O = \{o_1, o_2, \ldots, o_m\}$, Moving object set $V = \{v_1, v_2, \ldots, v_n\}$, Time domain $T = \{t_1, t_2, \ldots, t_h\}$, Check_in records $CK = \{\langle o_i, v_j, t_k \rangle \mid o_i \in O, v_j \in V, t_k \in T\}$, distance threshold $d$, time threshold $\tau$, visitor similarity threshold $min\_sim$;
**Output:** High visited co-location pattern set (SCPs);
% $CM_k$ is $k$-size candidates set; $C_2(c)$ is the 2-size subset of co-location $c$
**Begin**
1. Calculate spatial neighborhoods table containing visitor similarity
2. Divide spatial neighbor relationships into G-set and L-set by $min\_sim$;
3. Calculate and collect 2-size spatial high visited co-location patterns;
4. Let k = 3, generate $k$-size candidate patterns according to NIL and store them in $CM_k$
5. While $CM_k$! = NULL
　　　5.1 for each $c \in CM_k$
　　　　　5.1.1 If Optimization_Strategy_1(c, NIL) = True:
　　　　　　　Calculate minimum visitor similarity of $c$ $Mvs(c)$;
　　　　　　　If $Mvs(c) \geq min\_sim$, then $c$ is a spatial high visited co-location.
　　　　　　　pattern, which can be put into the set SCPs;
　　　　　　　if $Mvs(c) < min\_sim$, execute Optimization_Strategy_2(c, L, CMk);
　　　　　5.1.2 Else execute Optimization_Strategy_2(c, L, CMk);
　　　5.2 Generate k + 1-size candidates $CM_{k+1}$ based on CMk and NIL, k = k + 1;
6. Put out all high visited co-location patterns SCPs;
**End**

---

## 5. Experimental Result Analysis

*5.1. Experimental Datasets*

To evaluate the effectiveness of the EHVSCM algorithm, we use three real datasets for experiments, they are described in detail below:

**Data set 1**: This dataset includes about 12 months of check-in data in MESA (USA) collected from YELP from 3 January 2017 to 24 December 2017. There are 5736 spatial instances, 21 spatial features and 33337 visitors.

**Data set 2**: This dataset includes about ten months of check-in data in New York City (USA) collected from Foursquare from 12 April 2012 to 16 February 2013. There are 42981 spatial instances, 251 spatial features and 1038 visitors.

**Data set 3**: This dataset includes about ten months of check-in data in Tokyo (Japan) collected from Foursquare from 12 April 2012 to 16 February 2013. There are 67667 spatial instances, 247 spatial features and 2293 visitors.

**Download Link**: http://sites.google.com/site/yangdingqi/home/foursquare-dataset (accessed on 1 January 2016). https://www.yelp.com/dataset (accessed on 1 January 2019).

All experiments are compiled in Java and implemented using a PC with 96 GB of main memory and an i7 CPU.

*5.2. Rationality Analysis*

In this section, we conducted numerical analysis and similarity evaluation on traditional PI-based and MV-based algorithms.

- Numerical Analysis

Two measurement indices (PI and Mvs) have different ranges when considering the same dataset and distance threshold. In most cases, the traditional participation index is greater than the minimum visitor similarity. Table 3 records the extremum and average participation index and visitor similarity value for two different measurement indices. The maximum and average values of the participation index are all greater than those of the visitor similarity. This is because the PI mainly focuses on the independence of instances in the row instances of co-location, and these spatial relationships are static.

**Table 3.** Numerical analysis between PI and Mvs.

| | PI | | | Mvs | | |
|---|---|---|---|---|---|---|
| | **Minimum** | **Maximum** | **Average** | **Minimum** | **Maximum** | **Average** |
| Data set 1 | 0 | 0.887 | 0.2201 | 0 | 0.564 | 0.073 |
| Data set 2 | 0 | 0.937 | 0.325 | 0 | 0.5333 | 0.098 |
| Data set 3 | 0 | 0.9131 | 0.372 | 0 | 0.6201 | 0.104 |

Moreover, the visitor similarity mainly emphasizes the similarity of visitors between the spatial instances. The spatial instances don't change, but visitors can move anytime. The proximity in geographic space is always more stable than visitor similarity, so PI is always larger.

We also discussed the impact of distance threshold on minimum visitor similarity. As shown in Figure 5, we observed the changes in the mean and maximum values of Mvs by gradually increasing the distance threshold from 10 m to 50 m. The experimental results show that the changes in the average and maximum values of visitor similarity do not necessarily vary with distance. This is because the subjectivity of the visited information and the change in distance have little effect on the visitor similarity of co-location patterns.

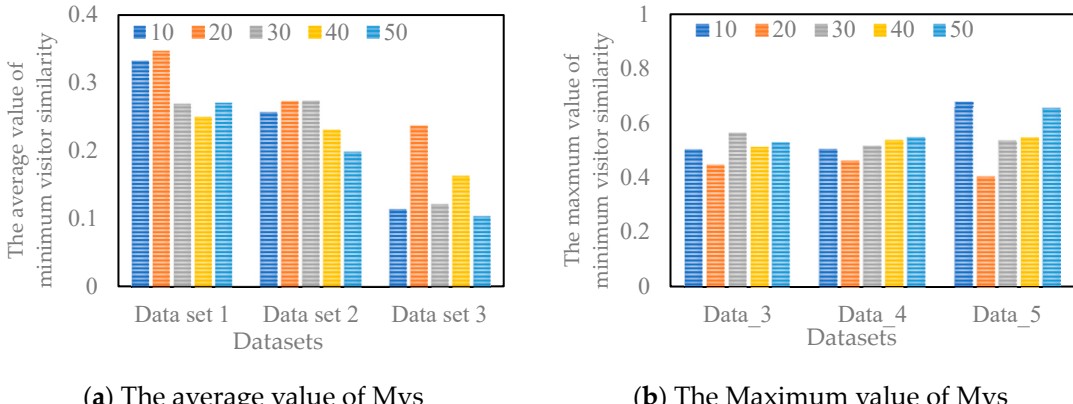

(**a**) The average value of Mvs　　　　　　(**b**) The Maximum value of Mvs

**Figure 5.** Discussion on the average and maximum values of minimum visitor similarity.

- Visitor similarity Evaluation

Starting from the mining purpose of highly visited co-location patterns, the visitor similarity reflected by the co-location patterns extracted by the ESHVCM should be greater than that of traditional frequent co-location patterns. Therefore, to verify the correctness of Mvs, we defined the "Similarity Ratio" to compare the visitor similarity of the top-*k* patterns mined by the ESHVCM algorithm (Mvs-based) and join-less algorithm(PI-based) [9], the specific calculation method for Similarity Ratio is as follows.

$$SR = \frac{\sum_{<o_i,o_j> \in Pneigh(CP)} vs(o_i,o_j)}{\sum_{o_i,o_j \in S \ \wedge dist(o_i,o_j)<d} vs(o_i,o_j)} \tag{7}$$

*CP* is a pattern set that contains the patterns to be evaluated, while *Pneigh*(*CP*) contains all neighboring instance pairs participating in the patterns in *CP*, *Pneigh*(*CP*) = {$<o_i, o_j>$ | $\exists c$ $\subseteq CP \wedge <o_i, o_j> \in insT_c$}. Therefore, $\sum_{<o_i,o_j> \in Pneigh(CP)} vs(o_i,o_j)$ is the sum of the visitor similarities of all neighboring instance pairs participating in the co-location patterns in *CP*. And $\sum_{o_i,o_j \in S' \wedge dist(o_i,o_j)<d} vs(o_i,o_j)$ is the sum of the visitor similarity of all neighboring instance pairs in the spatial dataset *S*, *SR* is the ratio of the above two values.

Figure 6a shows that the *SR* values of different sizes among the top 100 co-location patterns mined under two different metrics have been calculated. At this point, the evaluated co-location pattern set *CP* is the patterns of different sizes among the top 100 co-location patterns. The result indicates that the similarity ratio of the top-*k* mined by the PI-based method is always lower than the Mvs-based method in different datasets. From Figure 6a, it can be seen that by calculating the similarity ratio of the top-k co-location pattern under the two methods (*CP* pattern set consists of top-*k* co-location patterns under different k), it can be found that the similarity ratio of the top-*k* mined by the PI-based method is always lower than Mvs-based method. The experimental results verify that the high visited co-location pattern is superior to the traditional frequent co-location pattern regarding visitor similarity.

### 5.3. Quality Evaluation

In this section, like [27], we design a sample social entropy for measuring the quality of the co-locations based on visitor similarity. Firstly, we suppose *CP* = {$C_1$, $C_2$, ..., $C_n$} is the set of discovered co-location patterns, $U_{Ci}$ be the set of moving objects who visit co-location $C_i$, and let $U_{CP}$ be the set of moving objects who visit the spatial instances in *CP*. The social entropy of *CP* is then defined as:

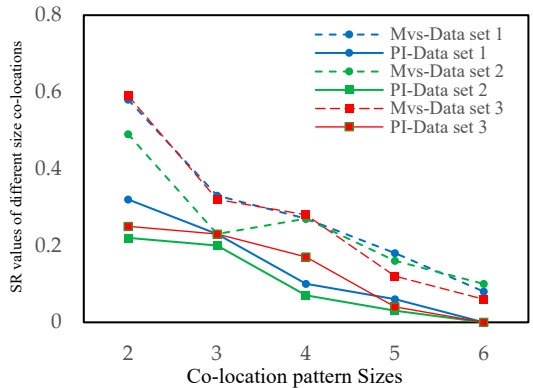

(**a**) SR values of different size co-locations

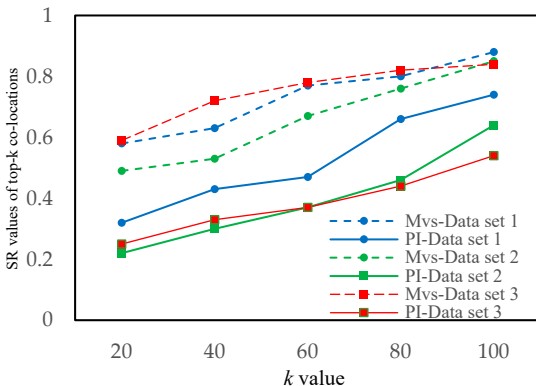

(**b**) SR values of top-*k* co-locations

**Figure 6.** Visitor similarity evaluation.

$$\varepsilon = \sum_{C_i \in CP} -\frac{|U_{C_i}|}{|U_{CP}|} \log \frac{|U_{C_i}|}{|U_{CP}|} \tag{8}$$

A lower social entropy means that the great majority of the visitors of *CP* visited the same co-location, indicating that these spatial instances in *CP* have tighter social connections.

We use social entropy to test our algorithm and Join-based algorithm based on PI from three aspects: different spatial distance thresholds, different time thresholds and different thresholds of participation index and social similarity.

1. *Changing Spatial Distance Threshold*. The spatial distance threshold is a crucial factor in constructing co-locations. This section mainly discusses the influence of different distance thresholds on the social entropies of co-location patterns. We set *min_prev* = 0.4, and *min_sim* = 0.3, $\tau$ = 5 days. Figure 7 shows the average social entropies of co-location patterns based on visitor similarity and PI in three datasets. With the increasing of spatial distance thresholds d from 100 to 350 m, the social entropies of the co-locations based on visitor similarity are always lower than the co-locations based on PI because the co-locations mined by our algorithm contain more social ties than the traditional co-locations which can help us mine more co-locations visited by similar visitors. Furthermore, as the spatial distance threshold increases, the number of neighborhood relationships and related mobile objects also increases since social entropies are increased with the increase of spatial distance thresholds.

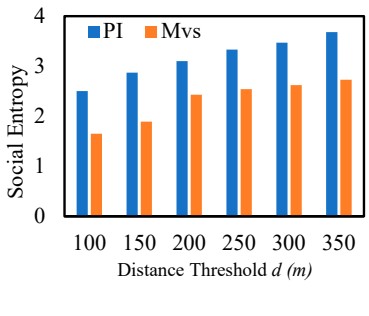

(**a**) Data set 1

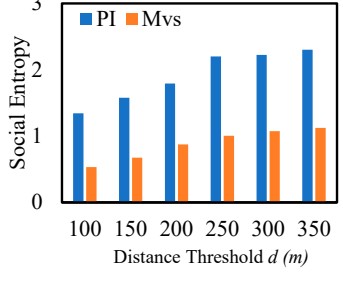

(**b**) Data set 2

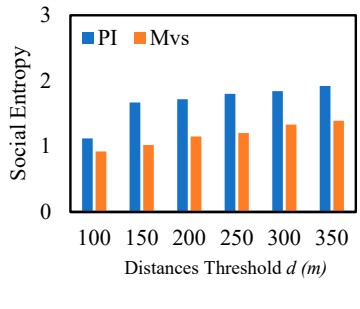

(**c**) Data set 3

**Figure 7.** Social entropy evaluation with different spatial distance thresholds.

2. *Changing time threshold*. When calculating the visitor similarity weight of the neighborhood, the time threshold is a very important factor. In this part, we change the time threshold to evaluate the social entropy of co-locations. For three data sets, we randomly selected two consecutive months of check-in records for experiments, and we

changed the time threshold from 10 days to 60 days. In Figure 8, we set *min_prev = 0.4*, *min_sim = 0.3*, *d = 200* m. With the increasing time thresholds, the social entropies of all co-locations are also increased because increasing the cardinality of visitors reduces the visitor similarity weight of the neighborhood. The social entropies of co-locations based on visitor similarity are always lower than the traditional co-locations in any datasets, which reflects the co-locations based on visitor similarity carry more social information and the spatial instances in co-locations are always visited by similar moving objects.

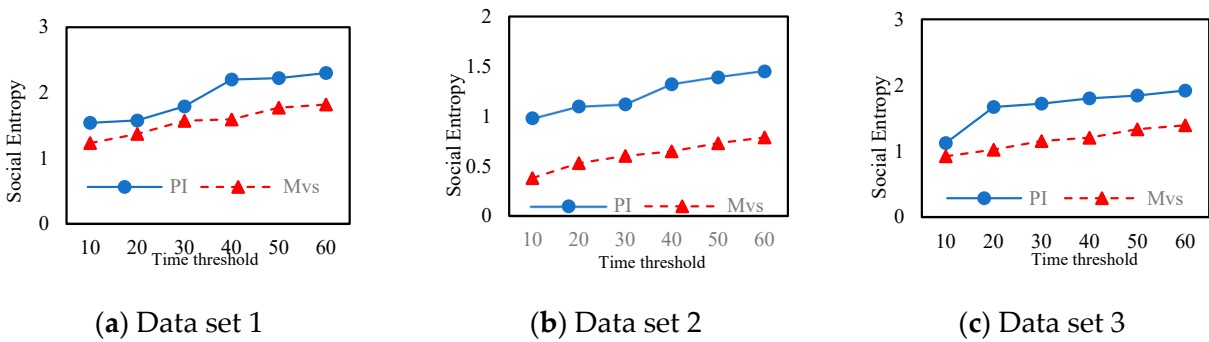

(**a**) Data set 1          (**b**) Data set 2          (**c**) Data set 3

**Figure 8.** Social entropy evaluation with different time thresholds.

3.  *Changing the thresholds of participation index and minimum visitor similarity*. In this experiment, we change the thresholds of PI and Mvs to evaluate the social entropies of discovered co-locations. We set the spatial distance threshold *d = 200* m, and we randomly selected ten days of check-in records from three datasets. For the co-locations based on PI, we change the threshold PI from 0.1 to 0.4. Figure 9 shows that the social entropies of co-locations are decreased with the increase of PIs. With the increase of PIs, the number of spatial instances that can participate in co-locations decreases, and the number of visitors who can participate in computing decreases. So, the proportion of similar visitors increases, and the social entropy decreases. For the co-locations based on visitor similarity, we change the threshold of visitor similarity from 0.1 to 0.4. From Figure 9, the social entropies of co-locations are also decreased with the increase of visitor similarity. The higher the visitor, the more similar moving objects visited the same co-location since the social entropies of co-locations are decreased with the increase of the threshold of visitor similarity. By evaluating social entropy, we can find that the co-locations based on visitor similarity have more social ties than the traditional co-locations.

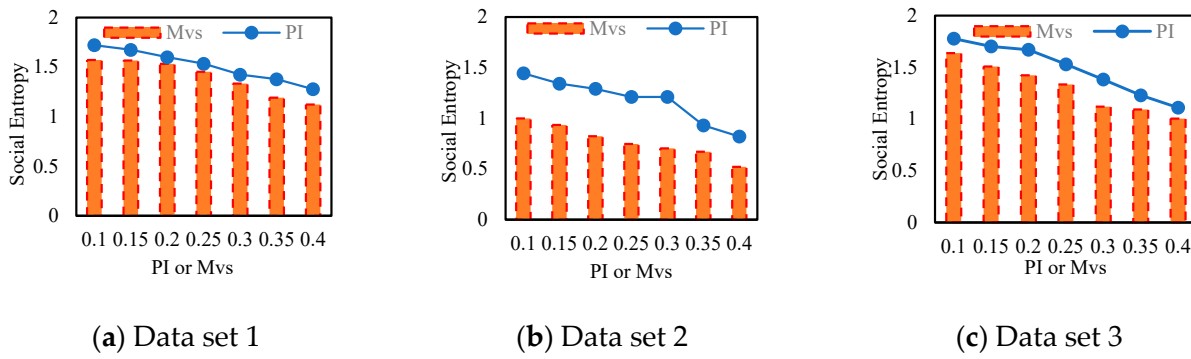

(**a**) Data set 1          (**b**) Data set 2          (**c**) Data set 3

**Figure 9.** Social entropy evaluation with different thresholds of participation index and minimum visitor similarity.

4.  *Comparing with other similarity indices*. We proposed visitor similarity weight of neighborhood to calculate the visitor similarity between two spatial instances, then the following measures also can be used:

*Jaccard.* The Jaccard index [27] is also called the Jaccard similarity coefficient, which can be used to compare the similarities and differences between finite sample sets. Given two spatial instances $o_i$ and $o_j$ with visitors sets $V_{oi}$ and $V_{oj}$. Jaccard similarity can be calculated as $J(o_i, o_j) = (|V_{oi} \cap V_{oj}|)/(V_{oi} \cup V_{oj})$ between the sets of visiting users of spatial instances $o_i$ and $o_j$, the social distance based on Jaccard can be defined as: $VS^{JAC}(o_i, o_j) = J(o_i, o_j)$, which focus on the similarity of visitors, and it don't consider the friendships between these users.

*Contribution users.* Compared with the Jaccard similarity, the method of Contribution User [27] considers the friendships between these visiting users. Given two spatial instances $o_i$ and $o_j$ with visitors sets $U_{oi}$ and $U_{oj}$, and a friendship set $E$, the contributing users set $CU_{ij}$ for the spatial instances $<o_i, o_j>$ is defined as:

$$CU_{ij} = \left\{ v_a \in U_{o_i} \middle| v_a \in U_{o_j} \text{ or } \exists\, v_b \in U_{o_j},\ (v_a, v_b) \in E \right\} \cup \left\{ v_a \in U_{o_j} | v_a \in U_{oi} \text{ or } \exists\, v_b \in U_{o_i},\ (v_a, v_b) \in E \right\}$$

Specifically, if a moving object $v_a$ has visited both $o_i$ and $o_j$, then $v_a$ is a contributing user. And if $v_a$ has visited spatial instances $o_i$, $v_b$ has visited spatial instances $o_j$, $v_a$ and $v_b$ are friends $(v_a, v_b) \in E$, both $v_a$ and $v_b$ are contributing users. According to $CU_{ij}$, we can define the visitor similarity as: $VS^{CU}(o_i, o_j) = \dfrac{|CU_{ij}|}{|U_{o_i} \cup U_{o_j}|}$.

Then, we experimented with Data set 1. Firstly, we change the spatial distance threshold from 200 to 700 m. Figure 10a shows that the social entropy of our visitor similarity is always better than that of Jaccard and Contribution users. All social entropies are increased by the increase of spatial distances. In Figure 10b, when changing the visitor similarity threshold, the performance of our visitor similarity is also much lower than the other two methods. This shows that our proposed method is more suitable for mining high visited co-locations. Because it considers both visiting time and the similarity between different visitors. And when calculating the user similarity between two spatial instances, our visitor similarity weight of the neighborhood first considers similar users rather than the same users, so its results are better than Jaccard similarity. Although Contribution users consider friendship among moving objects, the friendship does not mean that they are similar users, so in this paper, we use the moving objects' check-in history to evaluate the visitors' similarity, which is better than the results obtained by Contribution users.

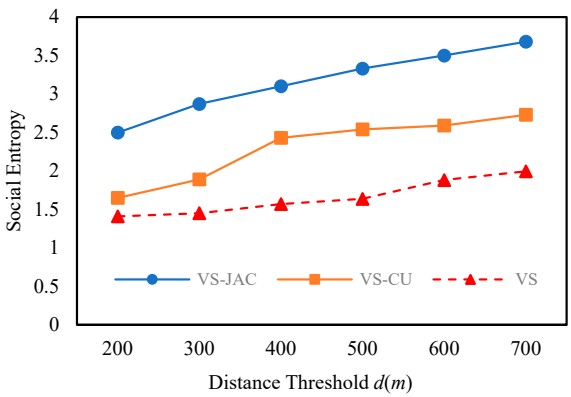

(**a**) Changing distance threshold

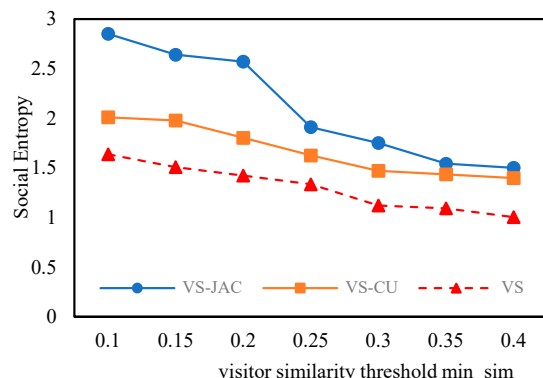

(**b**) Changing visitor similarity threshold

**Figure 10.** Comparing with other similarity indices.

### 5.4. Pruning Strategies Evaluation

In this paper, we propose two pruning strategies: table Instance checking and Visitor Similarity Weight Checking. This section evaluates the proposed pruning strategies regarding running time and pruning rate. We set the visitor similarity threshold *min_sim* = 0.3, time threshold $\tau$ = five days, and we randomly selected one month of check-in records for experiments from three datasets. In Figure 11, we compare the computational times

of the basic algorithm (BA) and the algorithm with pruning strategies. The running time of the latter is always less than that of the former. We also compare the running times of the algorithms proposed by this paper with the traditional mining methods, including join-based [1], join-less [9] and Fraction score [19]. Due to the addition of visited information calculation, the algorithm running time of BA is always higher than the traditional algorithms. After adding two pruning strategies, the running time of ESHVCM is already lower than the traditional algorithms. Additionally, we compared the runtime of algorithms ESHVCM, join-based, join-less and Fraction score on three datasets. As shown in Figure 12, with the distance threshold increased, the execution time of all four algorithms increased, but the execution time of ESHVCM was always lower than the other three algorithms.

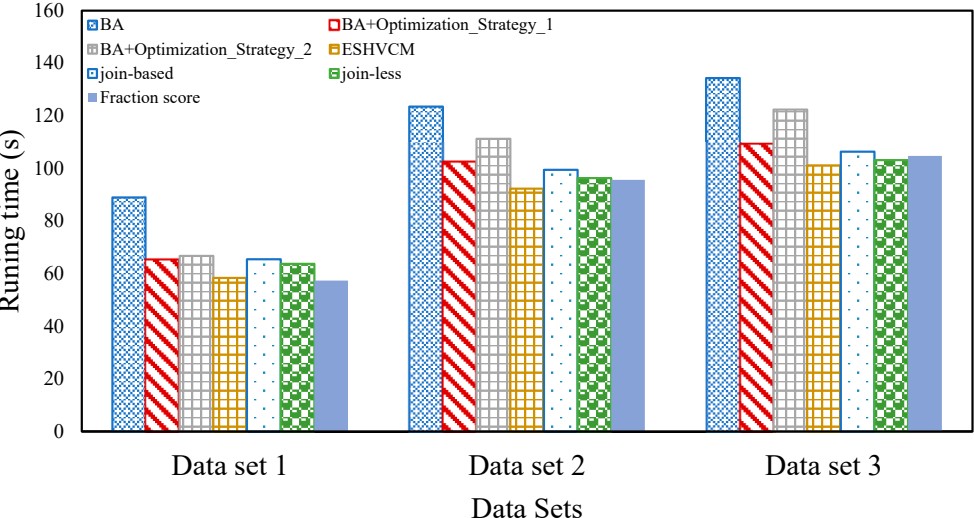

**Figure 11.** Running Times of different algorithms.

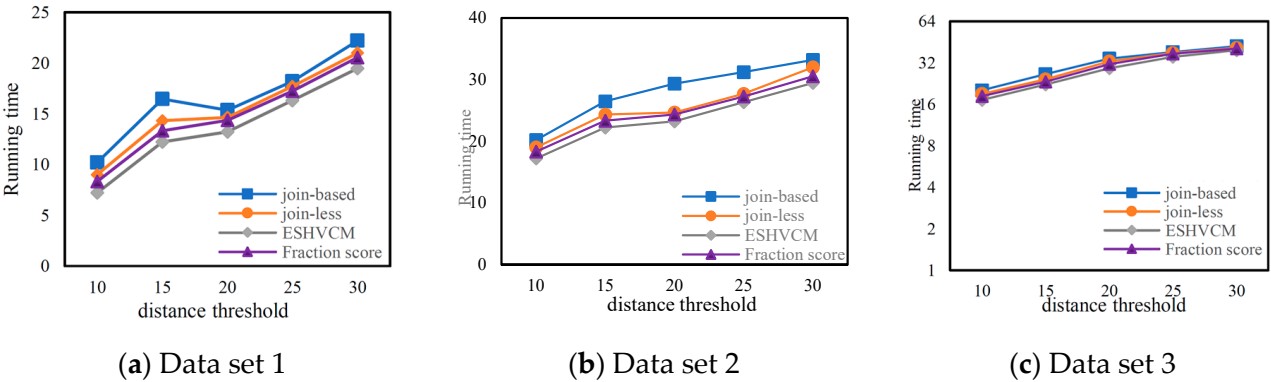

(**a**) Data set 1   (**b**) Data set 2   (**c**) Data set 3

**Figure 12.** Running Times of four algorithms on three datasets.

And in Figure 13, we calculate the pruning ratio to evaluate the efficiency of pruning strategies. The pruning ratio can be calculated as follows:

$$Pruning\ Ratio = \frac{the\ number\ of\ pruning\ candidates}{the\ number\ of\ all\ candidates}$$

Figure 13 shows that the pruning efficiency is the best when using table Instance checking and Visitor Similarity Weight Checking at the same time. Using Instance table checking and Visitor Similarity Weight Checking at the same time can prune the candidates from the spatial level and social level, which can filter out more co-location candidates, so it has the best pruning efficiency (the maximum pruning ratio in Figure 13 is 18.7%). In Data Set 1, the pruning efficiency is the worst when using Instance table checking only because

the relationships of spatial instances in Data Set 1 are very close, and pruning candidates from the spatial level is very hard. For Data Set 2 and Data Set 3, the existence of many social ties of the moving objects in these two datasets and pruning co-location candidates from the social level is very difficult because there are many visitor similarities are greater than *min_sim*, so the pruning efficiency is the worst when using Visitor Similarity Weight Checking only.

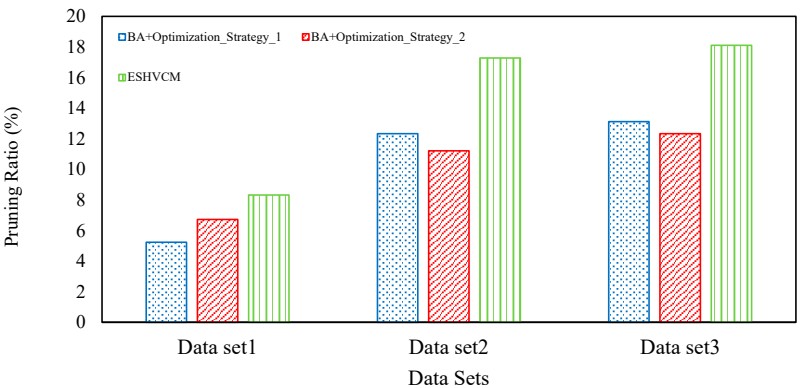

**Figure 13.** Pruning Ratio.

*5.5. Visualization Analysis*

In the last, we visually compared the co-location patterns found by the EHVSCM and Join-based algorithms.

Firstly, we chose the New York City check-in data in October 2012 from Data Set 2. We execute the EHVSCM and Join-based algorithms on this data and select several patterns from the mining results for analysis. Table 4 shows that the patterns <Residential Building, Train Station>, <Sporting Goods Shop, Gym/Fitness Center> have a very high participation index but very low visitor similarity. For these co-locations, they are very frequent adjacency in geographic space, which will disperse the moving objects, make the types of visitors very complex, and can't guarantee that they are often visited by similar users. The PI and Mvs of <Gym/Fitness Center, Coffee Shop> are very high, showing that its instances gather frequently in space and contain strong social relations. For another co-location pattern <Government Building, Cosmetics Shop>, the participation index and visitor similarity values are very close. This shows that the spatial instances of this pattern are not frequently adjacent, but they are constantly visited by stablers and similar moving objects. Compared with the first co-locations, this co-location reflects the social and spatial relations between spatial instances. Focusing on the last co-location pattern <Office, Bus Station, Pizza Place> in Table 4, it has a very low participation index, but its visitor similarity is not very low. Although their instances are not abundantly adjacent in geographic space, several neighboring spatial instances can always be visited by similar moving objects. This proves that the co-locations mined by our method consider spatial relationships and social connections between instances. Based on the above experimental analysis, we can conclude that the co-location patterns we mined are no longer a static neighborhood relationship between instances at the spatial level but endows it with more social information and relationships. The EHVSCM algorithm helps us find some more interesting patterns visited by similar visitors like <Office, Bus Station, Pizza Place> and <Government Building, Cosmetics Shop>.

In addition, we analyze the mining results of the three algorithms from the distribution graph of co-locations' instances and alternative approaches in the area of Queens in New York City. Figure 14a–c shows the co-locations by the EHVSCM algorithm (considering the spatial information and social similarity at the same time), Join-based algorithm (disregarding the social similarity between the spatial instances) and Visitor similarity (disregarding the spatial information).

**Table 4.** Mining results analysis of Data set 2.

| Co-Location Patterns in Data Set 2 | | |
|---|---|---|
| **Co-Locations** | **Participation Index** | **Minimum Visitor Similarity** |
| <Residential Building, Train Station> | 0.832 | 0.377 |
| <Sporting Goods Shop, Gym/Fitness Center> | 0.625 | 0.241 |
| <Gym/Fitness Center, Coffee Shop> | 0.725 | 0.438 |
| <Government Building, Cosmetics Shop> | 0.312 | 0.305 |
| <Office, Bus Station, Pizza Place> | 0.225 | 0.307 |

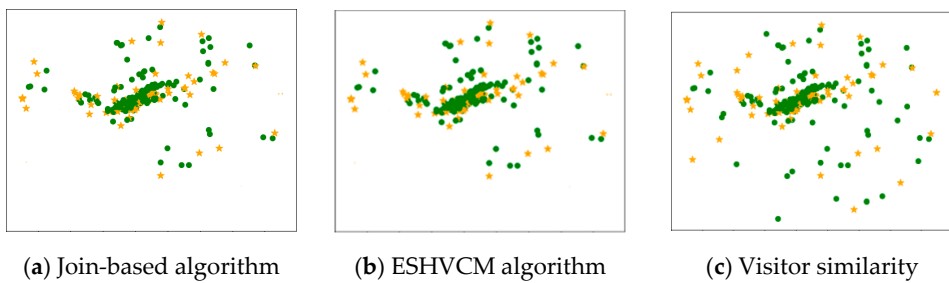

(**a**) Join-based algorithm     (**b**) ESHVCM algorithm     (**c**) Visitor similarity

**Figure 14.** Instance distribution of <Gym/Fitness Center, Park> with different algorithms. (Yellow star represents the Gym/Fitness Center and green Circle represents the Park).

We chose the co-location <Gym/Fitness Center, Park> to show its distribution graph by different algorithms. The introduction of minimum visitor similarity has an impact on the whole mining result set, but for a single co-location pattern, its PI remains unchanged in both the Join-less algorithm and EHVSCM algorithm, which is confirmed by the distributions of <Gym/Fitness Center, Park> are exactly alike in Figure 14a,b. In Figure 14c, only considering the visitor similarity between spatial instances, we can find that its distribution is more cluttered, and there is no clustering in the EHVSCM algorithm. It can be seen that the EHVSCM algorithm mining pattern with high visitor similarity without changing the degree of participation is obtained.

Thirdly, we chose three co-locations <Jewelry Store, Nail salon>, <Jewelry Store, Art Gallery> and <the History Museum, Art Gallery>. From Figure 15 (in this figure, red is the Art Gallery, green is the History Museum, blue is the Jewelry Store, and orange is the Nail salon), we can find that the number of Art galleries has decreased when using visitor similarity to measure the co-locations. This is because the co-location pattern <Jewelry Store, Art Gallery> is not a high visited co-location when considering the spatial information and social information at the same time, and the method we proposed helps us filter some co-location patterns with the lower social connection.

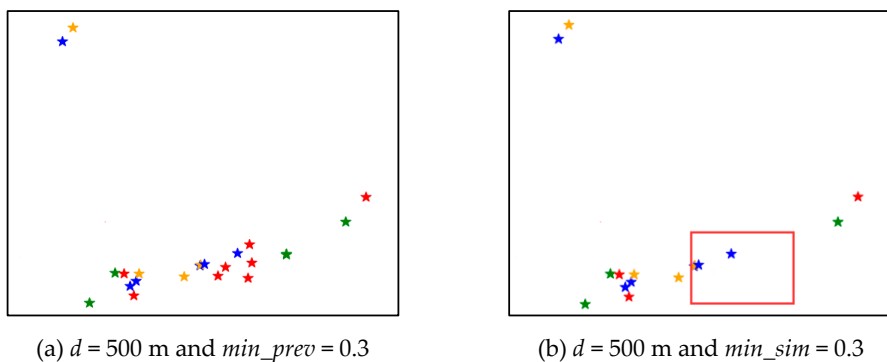

(a) *d* = 500 m and *min_prev* = 0.3        (b) *d* = 500 m and *min_sim* = 0.3

**Figure 15.** Instance distribution of <Jewelry Store, Nail salon>, <Jewelry Store, Art Gallery> and <History Museum, Art Gallery>.

## 6. Conclusions

This paper studies the problem of mining spatial high visited co-location patterns. Our mining framework extends the traditional spatial co-location pattern mining method by considering both the spatial neighborhood relationships and visiting information of spatial instances. We define a visitor similarity weight to evaluate the visitor similarity between spatial neighborhoods. Then, we use a new measure index minimum visitor similarity to reflect the social connections of the spatial features in the co-locations. Additionally, we propose a basic mining algorithm and design two pruning strategies from the spatial and social levels, which greatly reduce the time consumption caused by the calculation of social relations. In our experiments with three real datasets, we design and use the social entropy for measuring the social quality of the co-locations based on the visitor similarity, which confirms that the co-location patterns that we mined carry more social information and relationships than the traditional co-location patterns.

However, the mining framework we proposed in this paper also has room for further improvement. For example, due to the consideration of spatial relationships and social ties at the same time, it requires more computational time to mine the co-location patterns from spatial datasets with visiting information. Although visited time is considered in calculating visitor similarity between spatial instances, there is no specific analysis of the impact of time on visitor similarity. Thus, we will consider these problems in future studies.

**Author Contributions:** Conceptualization, X.W.; methodology, W.X. and X.W.; software, S.G. and X.W.; validation, W.X.; formal analysis, X.W.; investigation, S.G.; data curation, P.J.; writing—original draft preparation, X.W.; funding acquisition, W.X. All authors have read and agreed to the published version of the manuscript.

**Funding:** This research was funded by the National Natural Science Foundation of China under grant number 61862066, and the Yunnan Province Science Foundation under Grant No.202301AU070152.

**Data Availability Statement:** Publicly available datasets were analyzed in this study. This data can be found here: http://sites.google.com/site/yangdingqi/home/foursquare-dataset (accessed on 1 January 2014). https://www.yelp.com/dataset (access on 1 January 2019).

**Acknowledgments:** This research was supported by the Engineering Research Center of Computer Vision and Intelligent Control Technology, Department of Education of Yunnan Province.

**Conflicts of Interest:** The authors declare no conflict of interest.

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
