# Peer review of "Mining Highly Visited Co-Location Patterns Based on Minimum Visitor Similarity Constraints"

_electronics, doi:10.3390/electronics12183961_

Round 1
Reviewer 1 Report
Please find my comments in the attached report.

N/A
Reviewer 2 Report
In this paper, the authors propose a special type of co-location pattern “High visited co-location patterns”, which considers the spatial proximity and visitor similarity of spatial features simultaneously. And a new measurement “Minimum visitor similarity” has been proposed to reflect the visitor similarity of co-location patterns. Furthermore, by discussing the properties of the minimum visitor similarity, we propose an efficient algorithm to mine the highly visited co-locations and give two pruning strategies to improve the algorithm's efficiency.
The research topic is relevant. The research is well described and developed. The authors contextualized the study and the relevance of the theme. And the results were well presented.
I only suggest that the Authors improve the Introduction Section. I suggest that the authors include relevant citations to support this Section. There are several statements throughout this Section that are not referenced. Thus, after this correction, I suggest that the study be accepted.
Reviewer 3 Report
This paper proposes a special collaborative localization mode named High Access Collaborative Localization Mode and a new measurement method called Minimum Visitor Similarity to reflect visitor similarity in remote modes. At the same time, a pruning strategy is provided for algorithm optimization. Finally, the practicality and effectiveness of the proposed algorithm are demonstrated on real-world datasets. However, there are still some details in the paper that need to be revised. The details are as follows.
1. In abstract, we can see that the author has carried out experiments on three real data sets to prove the effectiveness of our proposed method. However, these real data sets are fuzzy. Please use specific data sets instead of real data sets in order to make readers read more intuitively.
2. In Section 2, the author analyzes the related work from two aspects, including the spatial co-location pattern mining methods and the analysis and mining methods based on social connections. Meanwhile, scholars' research results are analyzed, however, summary of the research results is ignored, please add it.
3. In Section 3.1, the definition of spatial co-location pattern mining is given. However, the meaning of characters in the definition is ignored. Please explain them.
4. In Section 4.3, an optimization strategy based on table instance checking in the previous section is proposed. By setting the visitor similarity threshold min_sim, the spatial neighbor relationship can be divided into two sets: G-set and L-set. Please explain how to determine the minimum threshold.
5. In Section 5, the author carries out experiments on three real data sets to prove the effectiveness of the proposed method. However, the download links of these data sets are ignored. Please provide the data download links for others to reproduce the experiment.
6. In Section 5.3, the author mentions the Jaccard method, however, the source of this method is ignored. Please add references about the source of Jaccard method.
7. According to the writing standard of the paper, all titles of figures in the paper should be centered. Please adjust the position of the title as required to make it more beautiful.
8. In this paper, the sizes of tables are different, please modify them.
9. Most of references are not state of the art, authors must use some more state-of-the-art papers. Few state of the art listed as follows.
â… ) https://doi.org/10.1109/TBDATA.2018.2871062
â…¡) https://doi.org/10.1109/TKDE.2019.2930598
â…¢) https://doi.org/10.1109/TEM.2021.3066090
â…£) https://doi.org/10.1109/TIP.2020.3015543
â…¤) https://doi.org/10.1016/j.bdr.2023.100369
Reviewer 4 Report
The paper suggests extending the traditional spatial co-location pattern mining method by taking into account both the spatial neighborhood relationships and visiting information of spatial instances. In addition, two pruning strategies are suggested: table Instance checking and Visitor Similarity Weight Checking.
1. In the related work section the authors cite many similar works. What is the novelty of their work? Why do not the authors put into the spotlight with more strength the potential of the method? Is the method novel enough? What are its differences with other similar methods? Why do they select this method and do not select other methods?
2. In the results section, what is "91GB of main member"? Did the authors mean "main memory"? Besides, 91GB seems an unreasonable size for main memory. The motherboard has two or four memory card slots where each memory card is a power of 2. Which combination of powers of 2 produced the result 91? This is impossible even with 4 slots.
3. The motivation and contribution of this paper should be further highlighted in the introduction.
4. The abstract needs a focus. The abstract of the manuscript does not clearly reflect the key content, as well as the reasons and effects of using this method.
5. The manuscript only emphasizes the advantages of the work done, but does not elaborate on the shortcomings of the work done and the follow-up prospects.
6. There is almost no explanation for Figure1. There is just one sentence in the text that refers to a small portion of the figure.
7. The format of references should be consistent. Sometimes the authors write Vol. and sometimes they do not. Also, the page numbers in reference no. 25 is written differently.
8. The problem definition should be given formally.
9. The core of the article is a summary of other articles. The overall contribution of the study and especially the pruning techniques is minor.
10. The proposed framework is not practical in a real-world scenario according to the manuscript's assumptions.
11. The equations and the results seem to be detached. Please explain how you have designed the experiments based on the theoretical background.
12. Equation 7 is a well-known equation but the authors present it as if they invented it and do not give any reference.
Round 2
Reviewer 1 Report
I would like to thank the authors for taking the time to address my comments in the previous review round.
However, certain issues that I raised earlier seem to remain and have not been addressed. Please find my comments/additional elaboration below.
1. Line 299 of the revised manuscript, as already pointed out in the previous batch of comments, the left-hand-side (LHS) and the right-hand-side (RHS) of the equations are exactly identical; as such, I am not sure what the goal is of presenting this equality -- which holds true at all times.
Additionally, I am confused about why the authors did not provide any explanation to this in their response and simply claimed that "there is no description of LHS and RHS in this paper", despite that they say "agreed and action taken".
2. Regarding the lack of benchmarking methods that was also pointed out in the previous review round. The authors responded that "due to the lack of many scholars proposing better methods in our research" -- I am not sure how to interpret this sentence.
Arguably, the proposed method provides an alternative to the existing ones by taking also into account the visit information along the time domain. However, to demonstrate the superiority of the proposed methodology, there are two aspects: (1) computational efficiency; (2) co-location pattern mining accuracy. The comparison against the "baseline" method is primarily to show (1), but currently, little is shown w.r.t. (2). As a matter of fact, the only benchmarking method considered is the join-base one, which is almost cherry-picking.
Therefore, I would suggest the authors to include more benchmarking methods -- that may or may not leverage visit information, to comprehensively show that (a) under what circumstances, ignoring the visit information is actually "okay" in the sense that existing methods can already given descent mining patterns; (b) under what circumstances, the proposed methodology dominates?; and (c) what are the strength and weakness of some existing methods and the proposed one, empirically?
I suggest the authors to go over their sentences carefully, as some of them read fairly awkward.
Reviewer 3 Report
The authors has made modifications according to the previous review comments, we recommend authors adding some latest references.
Minor editing of English language
Reviewer 4 Report
The authors made a decent effort and the paper is certainly publishable so I would recommend accepting the paper.
Round 3
Reviewer 1 Report
I thank the authors for taking the time and the effort to address the issues raised in the earlier review rounds. At this point, I don't have further comments.
N/A